# Transcription initiation at a consensus bacterial promoter proceeds via a 'bind-unwind-load-and-lock' mechanism

Abhishek Mazumder[1]*, Richard H Ebright[2], Achillefs N Kapanidis[1]*

[1]Biological Physics Research Group, Clarendon Laboratory, Department of Physics, University of Oxford, Oxford, United Kingdom; [2]Waksman Institute and Department of Chemistry, Rutgers University, Piscataway, United States

**Abstract** Transcription initiation starts with unwinding of promoter DNA by RNA polymerase (RNAP) to form a catalytically competent RNAP-promoter complex (RPo). Despite extensive study, the mechanism of promoter unwinding has remained unclear, in part due to the transient nature of intermediates on path to RPo. Here, using single-molecule unwinding-induced fluorescence enhancement to monitor promoter unwinding, and single-molecule fluorescence resonance energy transfer to monitor RNAP clamp conformation, we analyse RPo formation at a consensus bacterial core promoter. We find that the RNAP clamp is closed during promoter binding, remains closed during promoter unwinding, and then closes further, locking the unwound DNA in the RNAP active-centre cleft. Our work defines a new, 'bind-unwind-load-and-lock', model for the series of conformational changes occurring during promoter unwinding at a consensus bacterial promoter and provides the tools needed to examine the process in other organisms and at other promoters.

**\*For correspondence:**
abhishek.mazumder@physics.ox.
ac.uk (AM);
kapanidis@physics.ox.ac.uk (AK);
kapanidis@physics.ox.ac.uk
(ANK)

**Competing interest:** The authors declare that no competing interests exist.

## Introduction

Transcription initiation is the first and most highly regulated step in gene expression (*Mazumder and Kapanidis, 2019*; *Ruff et al., 2015*). During transcription initiation, RNA polymerase (RNAP), together with the transcription initiation factor σ, unwinds ~13 bp of promoter DNA to form a 'transcription bubble', and places the template-strand ssDNA of the unwound transcription bubble in contact with the RNAP active centre, yielding a catalytically competent RNAP-promoter transcription-initiation complex (RPo; *Mazumder and Kapanidis, 2019*). High-resolution structures of RPo define the contacts that RNAP and σ make with promoter DNA, as well as the conformation and interactions of unwound template-strand ssDNA engaging the RNAP active centre (*Bae et al., 2015*; *Zuo and Steitz, 2015*; *Zhang et al., 2012*). Structural and biochemical experiments suggest that transcription-bubble formation is initiated by unwinding the DNA base pair at the upstream end of the transcription bubble – breaking the base pair, and unstacking and flipping the non-template-strand base of the broken base pair, and inserting the unstacked and flipped non-template-strand base into a protein pocket of σ – followed by propagation of the unwinding in a downstream direction (*Mazumder and Kapanidis, 2019*; *Ruff et al., 2015*; *Zhang et al., 2012*; *Feklistov and Darst, 2011*). However, the mechanism by which DNA is unwound and loaded into the RNAP active-centre cleft has remained controversial (reviewed in *Mazumder and Kapanidis, 2019*). In crystal structures of RNAP σ holoenzyme, the RNAP active-centre cleft is too narrow to accommodate double-stranded DNA (dsDNA) (<20 Å), and σ obstructs access of dsDNA to the RNAP active-centre cleft (*Mekler et al., 2002*; *Murakami et al., 2002b*; *Murakami et al., 2002a*; *Vassylyev et al., 2002*). As a result, there is no unobstructed path by which dsDNA can access the RNAP active-centre cleft in RNAP holoenzyme (*Mekler et al., 2002*; *Murakami et al., 2002b*;

*Murakami et al., 2002a*; *Vassylyev et al., 2002*; *Young et al., 2002*; *Murakami and Darst, 2003*). Accordingly, it has remained unclear where, when, and how promoter DNA is unwound and loaded into the RNAP active-centre cleft.

To address these questions, two classes of models have been proposed. One class of models, termed 'load-unwind' models, propose that: (i) the RNAP active-centre cleft opens, through the swinging outward of one wall, termed the 'clamp', of the active-centre cleft, allowing loading of promoter DNA into the active-centre cleft as dsDNA; (ii) promoter DNA unwinds inside the active-centre cleft; and (iii) the active-centre cleft closes, through the swinging inward of the clamp, during or after DNA unwinding (*Mazumder and Kapanidis, 2019*; *Ruff et al., 2015*; *Mekler et al., 2002*; *Murakami et al., 2002b*; *Murakami et al., 2002a*; *Young et al., 2002*; *Darst et al., 2002*; *Chakraborty et al., 2012*). The other class of models, termed 'unwind-load' models, proposes that: (i) promoter DNA unwinds outside the RNAP active-centre cleft and (ii) unwound-promoter DNA loads into the active-centre cleft as single-stranded DNA (*Mazumder and Kapanidis, 2019*; *Vassylyev et al., 2002*). Some versions of the unwind-load model postulate that opening and closing of the RNAP clamp is required for DNA unwinding and DNA loading to occur (*Feklistov et al., 2017*; *Boyaci et al., 2019*). Other versions of the unwind-load model postulate that opening and closing motions of the RNAP clamp is not required for DNA unwinding and DNA loading (*Chen et al., 2020*).

Some previous results have been interpreted as supporting *load-unwind* models, including results from static DNA footprinting of trapped putative on-pathway intermediates in formation of RPo, suggesting the presence of dsDNA inside the active-centre cleft (*Spassky et al., 1985*; *Duval-Valentin and Ehrlich, 1986*; *Cowing et al., 1989*; *Schickor et al., 1990*), results from kinetic DNA footprinting suggesting the existence of on-pathway intermediates having dsDNA inside the active-centre cleft (*Davis et al., 2007*), fluorescence resonance energy transfer (FRET) results showing clamp opening and closing in RNAP and in trapped putative on-pathway intermediates in formation of RPo (*Chakraborty et al., 2012*; *Lin et al., 2018*; *Duchi et al., 2018*; *Mazumder et al., 2021*), functional correlations between inhibition of clamp opening and closing with inhibition of formation of RPo (*Chakraborty et al., 2012*; *Lin et al., 2018*; *Duchi et al., 2018*), and crystal and cryo-EM structures of trapped putative on-pathway intermediates in formation of RPo having dsDNA inside an open active-centre cleft (*Glyde et al., 2018*).

Other previous results have been interpreted as supporting *unwind-load* models, including time-resolved footprinting experiments suggesting that promoter unwinding occurs outside the active-centre cleft and precedes rate-limiting conformational changes in RNAP (*Rogozina et al., 2009*), and cryo-EM structures of trapped putative on-pathway intermediates in formation of RPo containing partly unwound DNA inside a closed active-centre cleft (*Chen et al., 2020*).

However, the previous results either have relied on analysis of artificially trapped complexes that have not been firmly established to correspond to *bona fide* on-pathway intermediates, or have relied on analysis of ensemble kinetics for which the identities and orders of appearance of intermediates have not been firmly established (*Chakraborty et al., 2012*; *Spassky et al., 1985*; *Duval-Valentin and Ehrlich, 1986*; *Cowing et al., 1989*; *Schickor et al., 1990*; *Davis et al., 2007*; *Lin et al., 2018*; *Duchi et al., 2018*; *Mazumder et al., 2021*; *Glyde et al., 2018*; *Rogozina et al., 2009*). Moreover, the previous results have been complicated by differences in the species source of the RNAP analysed, differences in the σ factors analysed, and differences in the promoter sequences analysed (*Chakraborty et al., 2012*; *Spassky et al., 1985*; *Duval-Valentin and Ehrlich, 1986*; *Cowing et al., 1989*; *Schickor et al., 1990*; *Davis et al., 2007*; *Lin et al., 2018*; *Duchi et al., 2018*; *Mazumder et al., 2021*; *Glyde et al., 2018*; *Rogozina et al., 2009*). Here, we report the use of single-molecule kinetic studies to define the pathway of DNA unwinding and DNA loading, without the assumptions that complicate analysis of artificially trapped complexes, and without the heterogeneity, population averaging, and time averaging that complicate analysis of ensemble kinetics. We used single-molecule promoter unwinding-induced fluorescence enhancement (smUIFE) to monitor DNA unwinding in solution in real time during formation of RPo, and we used single-molecule fluorescence resonance energy transfer (smFRET) to monitor opening and closing of the RNAP clamp in solution in real time during formation of RPo. In all experiments, we analysed *Escherichia coli* RNAP σ[70] holoenzyme at a consensus bacterial core promoter, comprising a consensus −35 element, a consensus −10 element, and a consensus −35/−10 spacer.

## Results

### Single-molecule UIFE

Previous work indicates that a promoter derivative having the fluorescent probe Cy3 site-specifically incorporated in the transcription-bubble region exhibits an ~2 -fold increase in fluorescence emission intensity upon promoter unwinding during RPo formation and exhibits an ~ 2 -fold decrease in fluorescence emission intensity upon promoter rewinding during promoter escape (*Feklistov et al., 2017*; *Ko and Heyduk, 2014*; *Koh et al., 2018*). These changes in fluorescence emission intensity provide a powerful approach to monitor promoter unwinding and rewinding in solution during transcription initiation (*Feklistov et al., 2017*; *Ko and Heyduk, 2014*; *Koh et al., 2018*). Here, we have adapted this approach to enable detection of promoter unwinding in solution at the single-molecule level in real time, and we designate our adaptation of the approach 'smUIFE', to underscore the similarity to the established method of single-molecule protein-induced fluorescence enhancement (*Hwang and Myong, 2014*; *Figure 1A*).

First, we analysed a promoter DNA fragment having Cy3 incorporated at a site at the downstream edge of the transcription bubble (non-template-strand position +2) of a consensus $\sigma^{70}$-dependent bacterial promoter (lacCONS-[+ 2 Cy3]; *Figure 1—figure supplement 1A*). Upon adding the Cy3-containing promoter DNA fragment to *E. coli* RNAP $\sigma^{70}$ holoenzyme immobilised on a coverslip mounted in a total-internal-reflection fluorescence (TIRF) microscope, we detected the appearance of fluorescence signal from single fluorescent species, indicating binding of single molecules of Cy3-containing promoter DNA fragment to surface-immobilised single molecules of RNAP holoenzyme (*Figure 1A*, bottom). Control experiments show that: (i) Cy3-containing promoter DNA fragments bound exclusively to immobilised RNAP molecules (*Figure 1—figure supplement 2A*), and the majority (~60%) of the resulting complexes were resistant to challenge with heparin (*Figure 1—figure supplement 2B*); (ii) binding of Cy3-containing non-promoter DNA fragments to immobilised RNAP holoenzyme molecules did not result in heparin-resistant complexes (*Figure 1—figure supplement 2C*); and (iii) binding of Cy3-containing promoter DNA fragments to immobilised RNAP core enzyme molecules did not result in heparin-resistant complexes (*Figure 1—figure supplement 2D*). These results confirm that sequence-specific, heparin-resistant, complexes were formed only between RNAP holoenzyme and Cy3-containing promoter DNA fragments.

We extracted intensity vs. time trajectories for the formation of RNAP-promoter complexes, and identified different classes of trajectories. A large class of trajectories (~45%) was characterised by the appearance of a fluorescence intensity of ~200 counts, followed by an increase in fluorescence intensity to ~450 counts, followed by either a decrease of the intensity to ~200 counts or a disappearance of the intensity (*Figure 1B*, top left; *Figure 1—figure supplement 3*, *middle*). Control experiments show that (i) binding of Cy3-containing non-promoter DNA fragments to immobilised RNAP holoenzyme did not result in ≥2 -fold fluorescence enhancement (*Figure 1—figure supplement 3C*); and (ii) binding of Cy3-containing promoter DNA fragments to immobilised RNAP core molecules did not result in ≥2 -fold fluorescence enhancement events for the overwhelming majority of the time trajectories (~96 %; *Figure 1—figure supplement 3D*). These results indicate that the detected ~2 -fold fluorescence enhancement events represent sequence-specific complexes between RNAP holoenzyme and promoter DNA fragments. Therefore, we assigned the states with no intensity as species that lack promoter DNA or that have promoter DNA with photobleached Cy3; the states with an intensity level of ~200 counts as RNAP-promoter complexes having dsDNA at the Cy3 incorporation site; and states with an intensity level ~450 counts as RNAP-promoter complexes having single-stranded DNA at the Cy3 incorporation site. This assignment yields a reaction sequence comprising: binding of double-stranded promoter DNA to RNAP (~200 counts; 'pre-unwinding state'), followed by promoter unwinding (~450 counts; 'unwound state'), followed by promoter rewinding (~200 counts) or probe photobleaching (~0 counts). Upon addition of an NTP subset enabling synthesis of transcripts up to 11 nt in length (ATP, UTP, and GTP), ~ 49 % of trajectories showed formation of a stable unwound-promoter state (with fluorescence intensity similar to that of the sub-population exhibiting promoter unwinding), as expected for the formation of initial transcribing complexes (RPitc ≤11; *Figure 1—figure supplement 3B*, top; see also Materials and methods). Upon subsequent addition of a separate NTP subset (GTP and CTP), enabling synthesis of transcripts up to 14 nt in length, ~ 27 % of the resulting complexes showed formation of a stable rewound-promoter state (*Figure 1—figure supplement 3B*, bottom), as expected for promoter escape and formation of a transcription elongation

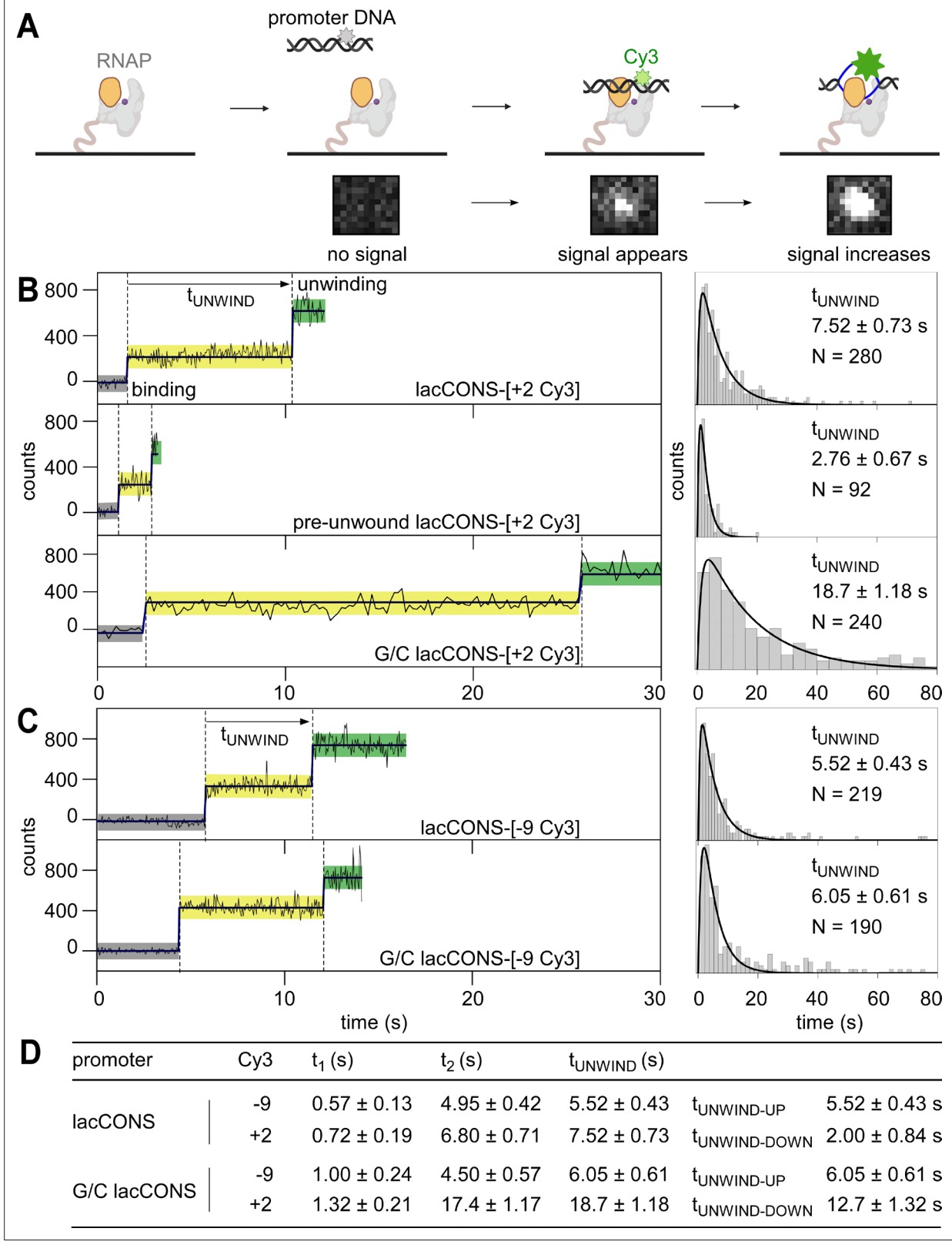

**Figure 1.** Single-molecule promoter unwinding-induced fluorescence enhancement (smUIFE): DNA unwinding in the upstream part of the transcription bubble precedes DNA unwinding in the downstream part of the transcription bubble. (**A**) (*Top*) Design of experiment monitoring promoter unwinding in real time. Grey, RNAP; orange, RNAP clamp; purple dot, RNAP active-centre; black, ds-DNA; blue, ss-DNA; light green, Cy3 on ds-DNA; dark green, Cy3 on ss-DNA. (*Bottom*) A cropped area (0.94 μm × 1.034 μm) of the field of view, showing appearance and enhancement of fluorescence signal from

*Figure 1 continued on next page*

*Figure 1 continued*

binding of single Cy3-labelled promoter fragment to an immobilised RNAP molecule. (**B**) (*Left*) Time trajectories of intensity from Cy3 on downstream segment of promoter bubble. Black, raw intensity; dark blue, idealised intensity; hidden Markov model (HMM)-assigned states: no promoter (black bars), closed promoter (light yellow bars) and open promoter (green bars). Frame rates: 50 ms, top and middle; 200 ms, bottom. Laser powers: 0.60 mW, top and middle; 0.15 mW, bottom. (*Right*) Dwell-time histograms of promoter state before unwinding, $t_{UNWIND}$. (**C**) (*Left*) Time trajectories of intensity from Cy3 on upstream segment of promoter bubble. Colours as in B. Frame rates: 50 ms. Laser powers: 0.6 mW. (*Right*) Dwell-time histograms of promoter state before unwinding, $t_{UNWIND}$. (**D**) Table comparing unwinding times for different promoter constructs.

The online version of this article includes the following source data and figure supplement(s) for figure 1:

**Source data 1.** Data for single-molecule UIFE experiments in *Figure 1*.

**Figure supplement 1.** Sequence of consensus lac-promoter fragments used in the study.

**Figure supplement 2.** Characterisation of complexes between RNAP and Cy3-labelled DNA fragments.

**Figure supplement 3.** Representative intensity vs. time trajectories from.

**Figure supplement 4.** Fluorescence intensities from Cy3-labelled promoter fragments pre- and post-unwinding.

complex. The results show that RNAP-promoter complexes formed in these experiments are transcriptionally active and are competent for promoter escape (see also Materials and methods).

Focusing on the class of trajectories that showed promoter unwinding after binding (as opposed to other classes of trajectories that did not show promoter unwinding after binding; *Figure 1—figure supplement 3A*), we analysed trajectories further to define the kinetics of promoter unwinding (see Materials and methods). Using hidden Markov model (HMM; *van de Meent et al., 2014*), we extracted dwell times for the pre-unwinding states and plotted a dwell-time histogram (*Figure 1B*, *top right*). The dwell-time histogram showed a peaked distribution, indicating a non-Markovian process having more than one rate-limiting steps before the formation of the unwound state (*Floyd et al., 2010*). We fitted this histogram to a two-exponential function (see Materials and methods for a detailed explanation; *Floyd et al., 2010*), and estimated the total time spent in the pre-unwinding state to be ~7.5 s (*Figure 1B*, *top right*; *Figure 1D*).

To assess whether the assay reports accurately on the kinetics of promoter unwinding, we analysed altered promoter derivatives predicted to unwind *more quickly* or *more slowly* than lac*CONS*. To accelerate unwinding, we lowered the energy barrier for unwinding by introducing a non-complementary sequence at positions –10 to –4 relative to the transcription start site (pre-unwound lacCONS-[+ 2 Cy3]; *Figure 1—figure supplement 1*). Time trajectories for pre-unwound lacCONS-[+ 2 Cy3] were qualitatively similar to those for lacCONS-[+ 2 Cy3] in terms of the intensity increase (*Figure 1B*, *middle left*, S4A) and the shape of the dwell-time histogram (*Figure 1B*, *middle right*); however, the trajectories showed significantly shorter dwell times in the pre-unwinding state (~2.8 s vs. ~7.5 s; *Figure 1B*, *middle right* vs. *top right*). To decelerate the process, we raised the energy barrier for unwinding by introducing a G/C-rich sequence at positions –4 to +1 relative to the transcription start site (G/C lacCONS-[+ 2 Cy3]; *Figure 1—figure supplement 1*). Time trajectories for G/C lacCONS-[+ 2 Cy3] showed significantly longer dwell times in the pre-unwinding state (~18.8 s vs. 7.5 s; *Figure 1B*, *bottom right* vs. *top right*; *Figure 1D*). Taken together, these results show that dwell times in the pre-unwinding state depend on energy barriers for promoter unwinding, consistent with expectation that the dwell times report on the kinetics of promoter unwinding.

Next, to determine whether promoter unwinding occurs in one step, or in more than one step, we assessed whether unwinding of the upstream half of the transcription bubble (positions –11 to –5) coincides with, or does not coincide with, unwinding of the downstream half of the transcription bubble (positions –4 to +2). To probe unwinding of the upstream half of *GC* promoter bubble, we performed analogous smUIFE experiments using promoter derivatives having Cy3 incorporated in the upstream half of the transcription bubble, at template-strand position –9 (lacCONS-[–9 Cy3] and G/C lacCONS-[–9 Cy3]; and *Figure 1—figure supplement 1*). The resulting trajectories were similar in terms of intensity increases (*Figure 1—figure supplement 4B*) and dwell-time-distribution shapes to those obtained with promoter DNA fragments having Cy3 incorporated in the downstream edge of the transcription bubble, but the dwell times in the pre-unwinding state were significantly shorter: ~ 5.5 s vs. ~7.5 s for lacCONS-[–9 Cy3] vs. lacCONS-[+ 2 Cy3] (*Figure 1C*, *top right* vs. *Figure 1B*, *top right*; *Figure 1D*) and ~6.0 s vs. ~18.8 s for G/C lacCONS-[–9 Cy3] vs. G/C lacCONS-[+ 2 Cy3] (*Figure 1C*, *bottom right* vs. *Figure 1B*, *bottom right*; 1D). We conclude that, for the promoters

analysed, unwinding of the upstream half of the transcription bubble occurs faster than unwinding of the downstream half of the transcription bubble, and we conclude that we can estimate from our data both the reaction time required for unwinding of the upstream half of the transcription bubble, $t_{UNWIND-UP}$ (from the lifetime of the pre-unwinding state when Cy3 is incorporated at position –9; ~ 5.5 s for lacCONS), and the reaction time required for the subsequent unwinding of the downstream half of the transcription bubble, $t_{UNWIND-DOWN}$ (from the difference in lifetimes between the pre-unwinding state when Cy3 is incorporated at position +2 and the pre-unwinding state when Cy3 is incorporated at position –9; ~ 2.0 s for lacCONS; *Figure 1D*). Our results confirm previous results (*Boyaci et al., 2019*; *Chen et al., 2020*; *Rogozina et al., 2009*; *Suh et al., 1993*; *Helmann and deHaseth, 1999*; *Lim et al., 2001*; *Auner et al., 2003*), indicating that promoter unwinding proceeds in a step-wise fashion, in an upstream-downstream direction, and suggest that the step for upstream unwinding is slower than downstream unwinding for lacCONS (~5.5 s vs. ~2.0 s), whereas the step for downstream unwinding is slower than upstream unwinding for G/C lacCONS (~6.0 s vs. ~13.0 s).

## smUIFE in the presence of an inhibitor that prevents RNAP clamp opening

To determine whether promoter unwinding is affected by preventing opening of the RNAP clamp, we repeated our experiments in presence of myxopyronin (Myx; *Chakraborty et al., 2012*; *Feklistov et al., 2017*; *Mukhopadhyay et al., 2008*; *Belogurov et al., 2009*; *Dulin et al., 2018*), an RNAP inhibitor that prevents RNAP clamp opening (*Chakraborty et al., 2012*; *Duchi et al., 2018*; *Mukhopadhyay et al., 2008*; state with an smFRET efficiency, E*~0.36 in *Figure 2—figure supplement 1D*)

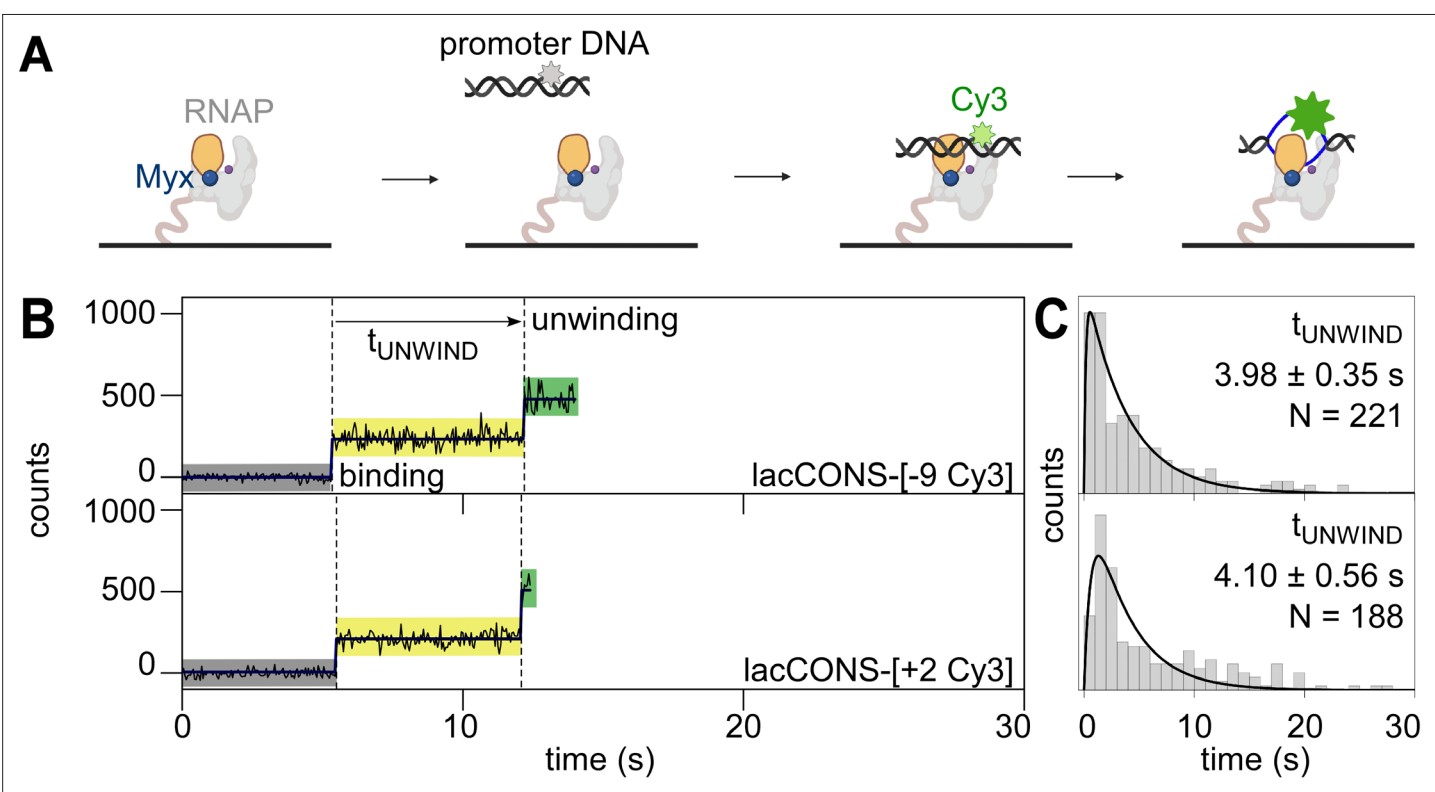

**Figure 2.** Single-molecule promoter unwinding-induced fluorescence enhancement (smUIFE) in the presence of an inhibitor that prevents RNAP clamp opening: preventing RNAP clamp opening does not prevent DNA unwinding. (**A**) Design of promoter unwinding experiment in presence of myxopyronin (Myx). Blue sphere, Myx; rest as in 1 A. (**B**) (*Left*) Time trajectory of intensity from Cy3 on upstream (*top*) and downstream (*bottom*) segment of promoter bubble. Colours as in 1B. Frame rates: 50 ms. Laser powers: 0.60 mW. (*Right*) Dwell-time histograms of promoter state before unwinding, $t_{UNWIND}$.

The online version of this article includes the following source data and figure supplement(s) for figure 2:

**Source data 1.** Data for single-molecule UIFE experiments in presence of Myxopyronin in *Figure 2*.

**Figure supplement 1.** Characterisation of complexes formed between RNAP and promoter fragments in presence of myxopyronin (Myx).

and that allows formation of heparin-sensitive RNAP-promoter complexes but prevents formation of heparin-resistant RNAP-promoter complexes (*Mukhopadhyay et al., 2008*; *Figure 2—figure supplement 1A–B*, ). We reasoned that, if clamp opening is obligatory for promoter unwinding, preventing clamp opening by addition of Myx should either prevent or delay downstream unwinding. We first performed smUIFE experiments in the presence of Myx using promoter DNA fragment lacCONS-[+ 2 Cy3], which has Cy3 incorporated at the downstream edge of the transcription bubble. The results showed a fluorescence intensity enhancement of ~2.4 -fold, consistent with unwinding of the downstream edge of the transcription bubble (*Figure 2B*, *bottom*; *Figure 2—figure supplement 1C*; see *Feklistov et al., 2017*) and showed a dwell time between initial binding and intensity enhancement of ~4.1 s (*Figure 2C*, *bottom*). We next performed smUIFE experiments in the presence of Myx using promoter DNA fragment lacCONS-[–9 Cy3], which has Cy3 incorporated in the upstream half of the transcription bubble. The results were essentially identical: a fluorescence intensity enhancement of ~2.4 -fold (*Figure 2B*, *top*; *Figure 2—figure supplement 1C*, *top*) and a dwell time between initial binding and intensity enhancement of ~4.0 s (*Figure 2C*). For both promoter derivatives, the observed fluorescence intensity enhancements were similar in the absence and presence of Myx (*Figure 1—figure supplement 4* vs. *Figure 2—figure supplement 1C*), and the observed dwell times between initial binding and intensity enhancement times were shorter – not longer – in the presence of Myx than in the absence of Myx (~3.5 s shorter for transcription-bubble downstream edge and ~1.5 s shorter for transcription-bubble upstream half; *Figure 2B–C* vs. *Figure 1B–C*, *top panels*). We suggest that Myx prevents formation of heparin-resistant open complexes either by preventing loading of unwound DNA into active-centre cleft or by preventing subsequent locking of the RNAP clamp. We infer – contrary to the models in which clamp opening is obligatory for promoter unwinding – that preventing RNAP clamp opening does not prevent promoter unwinding, and increases, not decreases, the kinetics of promoter unwinding.

## Single-molecule FRET

To assess *directly* whether RNAP clamp motions occur during RPo formation, we performed smFRET measurements using an RNAP derivative containing Cy3B, serving as a fluorescence donor, incorporated at the tip of the RNAP clamp and Alexa647, serving as a fluorescence acceptor, incorporated at the tip of the opposite wall of the RNAP active-centre cleft, monitoring smFRET efficiency (E*) during RPo formation (*Figure 3A*). In previous smFRET studies using the same probes, we showed that the RNAP clamp interconverts between open (E* ~ 0.2), partly-closed (E* ~ 0.3), and closed (E* ~ 0.4) conformations in solution (*Chakraborty et al., 2012*; *Lin et al., 2018*; *Duchi et al., 2018*; *Mazumder et al., 2021*). To monitor RNAP clamp motions during the formation of RPo, we immobilised promoter DNA molecules (biotin-lacCONS; *Figure 1—figure supplement 1*, *Figure 3—figure supplement 1*) on a coverslip mounted on a TIRF microscope, started recording, and added the doubly labelled RNAP (*Figure 3A*). Binding of doubly labelled RNAP molecules to DNA was monitored by detecting the simultaneous appearance of donor and acceptor fluorescence emission signals on the surface, and RNAP clamp motions in RNAP-promoter complexes at and after binding were monitored by quantifying E* (*Figure 3—figure supplement 2A*). Control experiments show that: (i) doubly labelled RNAP molecules bound exclusively to immobilised promoter DNA molecules (*Figure 3—figure supplement 1A*); (ii) the majority (~87%) of the resulting complexes were resistant to challenge with heparin (*Figure 3—figure supplement 1B*); and (iii) doubly labelled RNAP molecules did not bind stably to immobilised non-promoter DNA molecules (*Figure 3—figure supplement 2C*). These results indicate that sequence-specific, heparin-resistant RNAP-promoter complexes were only formed between the doubly labelled RNAP holoenzyme molecules and promoter DNA fragments.

In order to determine the RNAP clamp conformations immediately upon initial binding of RNAP to promoter DNA, we plotted distributions of E* values for the first five frames (0.5 s) after initial binding. The resulting distributions could be fitted to a Gaussian function with mean E* of ~0.4, indicating that the initial binding of RNAP to promoter DNA involved RNAP with a closed clamp (*Figure 3—figure supplement 2B*). Next, we examined full E* time trajectories, for up to ~60 s, following initial binding, seeking E* changes potentially consistent with RNAP clamp opening. We detected no E* changes – not even transient E* changes, within the ~100 ms temporal resolution of the analysis – that potentially could be assigned as consistent with clamp opening (E* of ~0.2; *Figure 3B*, area highlighted in cyan; *Figure 3—figure supplement 2A*).

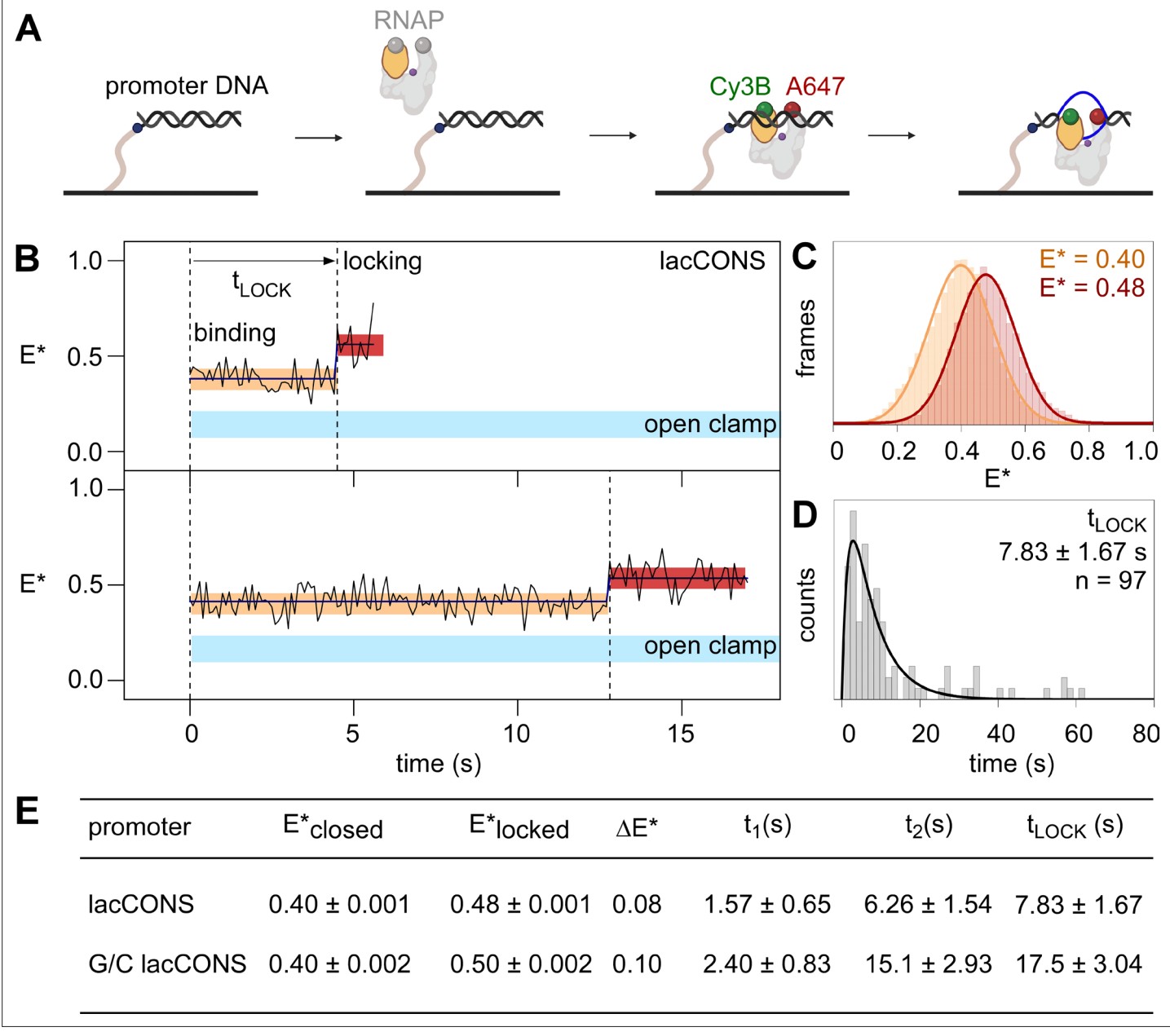

**Figure 3.** Single-molecule fluorescence resonance energy transfer (smFRET): DNA unwinding occurs without RNAP clamp opening and is followed by RNAP clamp locking. (**A**) Design of experiment monitoring clamp status in real time. Black, ds-DNA; orange, RNAP clamp; grey, rest of RNAP; purple dot, RNAP active-centre; blue, ss-DNA; green, Cy3B; and red, Alexa647. (**B**) Representative time trajectories of E* for experiments with a lacCONS promoter fragment, showing hidden Markov model (HMM)-assigned closed-clamp state (orange), locked-clamp state (red), and interstate transition (dark-blue). Expected range of E* values for an open-clamp state is highlighted in light blue. Frame rate: 100 ms. Laser powers: 200 μW in red and 500 μW in green. (**C**) HMM-assigned histograms and Gaussian fits of E* for full-time trajectories from experiments with a lacCONS promoter fragment. (**D**) Dwell-time histograms of time before transition to the locked-clamp state, $t_{LOCK}$, for experiments with a lacCONS promoter fragment. (**E**) Table showing mean E*; difference in E* (ΔE*) between closed-clamp or locked-clamp states and time to transition to a locked-clamp state after initial binding for the lacCONS and lacCONS-GC promoter fragments.

The online version of this article includes the following source data and figure supplement(s) for figure 3:

**Source data 1.** Data for single-molecule FRET experiments in *Figure 3*.

**Figure supplement 1.** Characterisation of complexes formed between clamp-labelled RNAP and immobilised promoter fragments.

**Figure supplement 2.** Single-molecule fluorescence resonance energy transfer (smFRET): initial binding of RNAP to surface-immobilised promoter DNA fragments take place via a closed-clamp conformation.

*Figure 3 continued on next page*

*Figure 3 continued*

**Figure supplement 3.** Classification for different time trajectories of E* of clamp-labelled RNAP molecules bound to immobilised lacCONS-promoter fragments in real time.

**Figure supplement 4.** Single-molecule fluorescence resonance energy transfer (smFRET) data showing binding of clamp-labelled RNAP to immobilised biotin-G/C lacCONS promoter fragments.

Next, we examined full E* time trajectories, following initial binding, seeking any E* changes potentially consistent with any RNAP clamp motions (see Materials ad methods and *Figure 3—figure supplement 3* for full classification). Intriguingly, a large fraction of trajectories (~44%) started at E*~0.40, indicative of the previously defined closed-clamp state, then transitioned to E*~0.48, indicative of a new, more tightly closed, clamp state, and then returned to E*~0.40 or photobleached (*Figure 3B–C*; *Figure 3—figure supplement 3B*, *top*). We refer to the new, more tightly closed, clamp state with E*~0.48, as the 'locked-clamp' state. We further analysed E* time trajectories to determine the time between initial binding of RNAP to promoter DNA and appearance of the locked-clamp state. The corresponding dwell-time histogram showed a peaked distribution and was fitted to a two-exponential function, yielding ~7.8 s, as the time between initial binding of RNAP to promoter DNA and appearance of the locked-clamp state (*Figure 3D*), a time that, within error, is identical to the ~7.5 s time between initial binding and unwinding of the downstream half of the transcription bubble.

Analogous experiments with a promoter derivative having G/C-rich sequence at positions –4 to +1 relative to the transcription start site, G/C lacCONS, yielded similar results: that is, initial binding by RNAP with a closed-clamp state (E*~0.41; *Figure 3—figure supplement 4A*; B, *left*); no trajectories showing transitions – not even transient transitions, within the ~400 ms temporal resolution of the analysis – to an open-clamp state during a period up to ~200 s after initial binding; a large fraction of trajectories (~48%), showing a transition to a locked-clamped state (E*~0.50, *Figure 3—figure supplement 4A*; B, *right*), and a time between initial binding and appearance of the locked-clamped state matching the time required for unwinding of the downstream half of the transcription bubble (*Figure 3E*, *Figure 3—figure supplement 4C*).

We conclude that the RNAP clamp is in a closed state upon promoter binding, that the RNAP clamp does not open--not even transiently, within the temporal resolution of our analysis – between promoter binding and promoter unwinding, and that the RNAP clamp closes further – 'locks' – after promoter unwinding.

## Discussion

Taken together, our results indicate that RPo formation by *E. coli* RNAP σ70 holoenzyme at a consensus bacterial core promoter proceeds through a 'bind-unwind-load-and-lock' mechanism, in which the RNAP clamp is closed upon promoter binding, remains closed during unwinding of promoter DNA

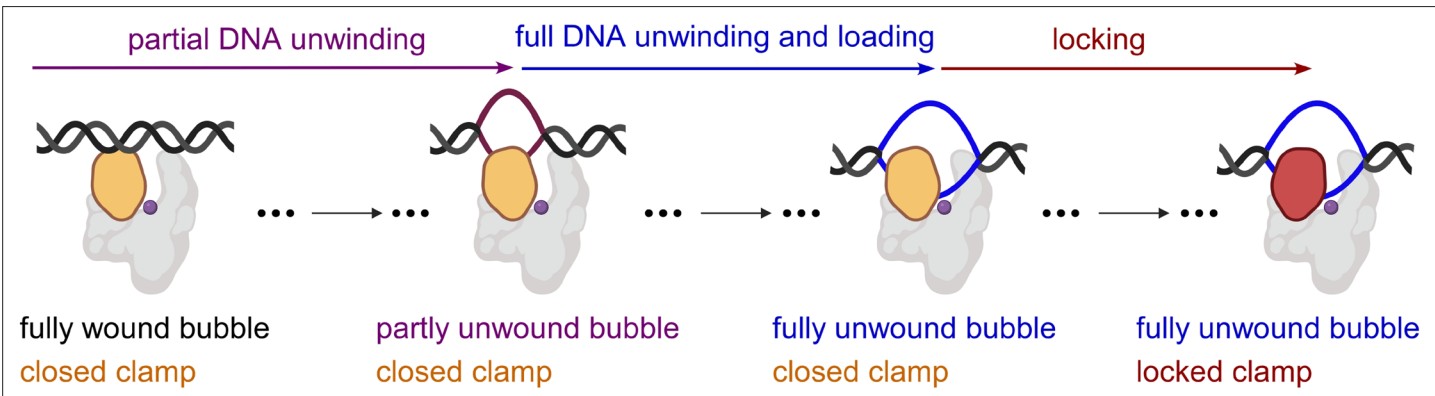

**Figure 4.** 'Bind-unwind-load-and-lock' mechanism for the formation of RPo. Orange, closed clamp; red, locked clamp; grey, rest of RNAP; purple dot, RNAP active-centre; black, ds-DNA; magenta, partly unwound bubble and blue, fully unwound bubble.

– which proceeds in an upstream-to-downstream direction – and then closes further, locking the unwound DNA in the RNAP active-centre cleft (*Figure 4*).

As described in the introduction, it has been the predominant view that structural considerations necessitate RNAP clamp opening for loading of dsDNA into the RNAP active-centre cleft but do not necessitate RNAP clamp opening for loading of ssDNA into the RNAP active-centre cleft (*Mazumder and Kapanidis, 2019*; *Ruff et al., 2015*; *Mekler et al., 2002*; *Murakami et al., 2002b*; *Murakami et al., 2002a*; *Young et al., 2002*; *Darst et al., 2002*; *Chakraborty et al., 2012*). Under that predominant view, our finding that the RNAP clamp remains closed throughout promoter unwinding at a consensus bacterial promoter implies that it is ssDNA, not dsDNA, that enters the RNAP active-centre-cleft, and thus that promoter unwinding occurs at least in part outside, not inside, the RNAP active-centre cleft (*Figure 4*).

In contrast to the predominant view, it recently has been speculated, based on molecular-dynamics simulations, that part of the RNAP cleft potentially could become accessible to both dsDNA and ssDNA without clamp opening, due to motions, putatively occurring on the nanosecond time scale, of the RNAP β lobe ('gate opening'; *Unarta et al., 2021*). Under this alternative view, our finding that the RNAP clamp remains closed throughout promoter unwinding at a consensus bacterial promoter potentially could be consistent with either ssDNA or dsDNA entering the RNAP active-centre cleft, and thus potentially could be consistent with promoter unwinding that occurs either outside or inside the RNAP active-centre cleft. Nevertheless, in view of the absence of experimental support for the alternative view, and in view of the mismatch between the nanosecond time scale of the limiting reaction in the alternative view and the millisecond-to-second time scales of promoter unwinding, we tend to disfavour this alternative view.

The consensus bacterial promoter employed in this study has been used in previous work to define transcription mechanisms, including RNAP clamp closure upon RPo formation (*Chakraborty et al., 2012*), DNA scrunching in transcription-start-site selection (*Winkelman et al., 2016*), DNA scrunching in initial transcription (*Kapanidis et al., 2006*), and RNAP pausing in initial transcription (*Duchi et al., 2016*; *Winkelman et al., 2020*), all of which subsequently were validated for other promoters. Nevertheless, we emphasise that it remains to be determined whether the bind-unwind-load-and-lock pathway defined here for the consensus bacterial promoter is also used by *E. coli* RNAP σ70 holoenzyme at other promoters and by other *E. coli* RNAP holoenzymes. Consistent with the possibility that a similar mechanism is used by *E. coli* RNAP σ70 holoenzyme at other promoters, a recent cryo-EM study of transcription initiation by *E. coli* RNAP σ70 holoenzyme at another promoter, *rpsT* P2, identified structural states having a partly unwound transcription bubble and a closed RNAP clamp (*Chen et al., 2020*). We note that the smUIFE and smFRET methods reported in this work could be applied to analyse transcription initiation by any *E. coli* RNAP holoenzyme at any promoter and potentially could be adapted to analyse transcription initiation by any RNAP – bacterial, archaeal, or eukaryotic – at any promoter. Extension of these assays using more complex labelling and imaging schemes should allow simultaneous observation of promoter unwinding and RNAP clamp conformational changes in the same molecule, enabling a more detailed view of the coupling of the molecular motions involved in promoter unwinding. Finally, development of massively parallel single-molecule assays interrogating large libraries of promoter sequences would enable a comprehensive analysis of general and specific features of this first step in transcription initiation for different promoter sequences.

# Materials and methods
## Preparation of reagents: RNAP, oligodeoxyribonucleotides, Myx
### RNAP derivatives

For experiments in *Figures 1 and 2*, hexahistidine-tagged *E. coli* RNAP holoenzyme was prepared using co-expression of genes encoding RNAP β', β, α, ω, and σ70 subunits to afford an RNAP σ70 holoenzyme derivative as follows: single colonies of *E. coli* strain BL21(DE3) (Millipore) co-transformed with plasmid pV10 (*Belogurov et al., 2007*) and plasmid pRSFduet-sigma (*Hudson et al., 2009*) were used to inoculate 20 ml LB broth (*Sambrook and Russell, 2001*) containing 100 µg/ml ampicillin and 50 µg/ml kanamycin and cultures were incubated 16 hr at 37°C with shaking. Culture aliquots (2 × 10 ml) were used to inoculate LB broth (2 × 1 L) containing 100 µg/ml ampicillin and 50 µg/ml kanamycin; cultures were incubated at 37°C with shaking until $OD_{600}$ = 0.6; IPTG was added to 1 mM; and

cultures were further incubated 3.5 hr at 37°C with shaking. Cells were harvested by centrifugation (4000 × $g$; 20 min at 4°C), re-suspended in 20 ml buffer A (10 mM Tris-HCl, pH 7.9, 200 mM NaCl, and 5% glycerol), and lysed using an EmulsiFlex-C5 cell disrupter (Avestin). The lysate was cleared by centrifugation (20,000 × $g$; 30 min at 4°C), precipitated with polyethyleneimine (Sigma-Aldrich) as in *Niu et al., 1996*, and precipitated with ammonium sulphate as in *Niu et al., 1996*. The precipitate was dissolved in 30 ml buffer A and loaded onto a 5 ml column of Ni-NTA-agarose (Qiagen) pre-equilibrated in buffer A; the column was washed with 50 ml buffer A containing 10 mM imidazole, and eluted with 25 ml buffer A containing 200 mM imidazole. The sample was further purified by anion-exchange chromatography on Mono Q 10/100 GL (GE Healthcare; 160 ml linear gradient of 300–500 mM NaCl in 10 mM Tris-HCl, pH 7.9, 0.1 mM EDTA, and 5% glycerol; flow rate = 2 ml/min). Fractions containing hexahistidine-tagged *E. coli* RNAP σ70 holoenzyme were pooled, concentrated to ~2 mg/ml using 30 kDa MWCO Amicon Ultra-15 centrifugal ultrafilters (EMD Millipore), and stored in aliquots at –80°C.

For experiments in *Figure 3* and *Figure 3—figure supplements 1–4*, fluorescently labelled, hexahistidine-tagged *E. coli* RNAP holoenzyme (hereafter 'labelled-RNAP') with Cy3B and Alexa647 at positions 284 on the β' subunit, and 106 on the β subunit, respectively, was prepared using in vivo reconstitution methods as described (*Lin et al., 2018*).

### Nucleic acids

Oligodeoxyribonucleotides were purchased from IBA Lifesciences, dissolved in nuclease-free water (Ambion) to a final concentration of 100 µM and stored at –20°C. Oligodeoxyribonucleotides used in *Figure 1*, *Figure 1—figure supplements 2–4*, and *Figure 2—figure supplement 1* were labelled with Cy3 *N*-hydroxysuccinimidyl ester (Fisher Scientific) as described (*Mukhopadhyay et al., 2003*). Oligodeoxyribonucleotides were annealed by mixing two complementary strands in a ratio of 1:1 in hybridisation buffer (50 mM Tris–HCl pH 8.0, 500 mM NaCl, 1 mM EDTA) and by heating for 5 min at 95°C, followed by cooling to 25°C in 2°C steps with 1 min per step using a thermal cycler (Applied Biosystems).

Myx was prepared as described (*Mukhopadhyay et al., 2008*).

### Single-molecule fluorescence microscopy: smUIFE experiments

For experiments in *Figures 1 and 2*, *Figure 1—figure supplements 2–4*, and *Figure 2—figure supplement 1*, observation wells for real-time experiments were prepared as described (*Lin et al., 2018*). Briefly, a biotin-PEG-passivated glass surface was prepared, functionalised with Neutravidin (Sigma-Aldrich), and treated with biotinylated anti-hexahistidine monoclonal antibody (Penta-His Biotin Conjugate; Qiagen), yielding wells with (biotinylated anti-hexahistidine monoclonal antibody)-Neutravidin-biotin-PEG-functionalised glass floors. Hexahistidine-tagged RNAP σ70 holoenzyme molecules were immobilised in observation wells with (biotinylated anti-hexahistidine monoclonal antibody)-Neutravidin-biotin-PEG-functionalised glass floors, as follows: aliquots (30 µl) of 0.1 nM hexahistidine-tagged RNAP σ70 holoenzyme in KG7 buffer (40 mM HEPES-NaOH, pH 7.0, 100 mM potassium glutamate, 10 mM MgCl₂, 1 mM dithiothreitol, 100 µg/ml bovine serum albumin, and 5 % glycerol) were added to the observation chamber and incubated 2–4 min at 22 °C, solutions were removed, wells were washed with 2 × 30 µl KG7, and 30 µl KG7 imaging buffer (KG7 buffer containing 2 mM Trolox, 1 mg/ml glucose oxidase, 40 µg/ml catalase, and 1.4% w/v D-glucose) at 22 °C was added.

For experiments in *Figure 1—figure supplement 2*, Cy3-labelled promoter (or non-promoter) fragments were manually added (final concentration of 2 nM) to the observation wells containing immobilised RNAP σ70 holoenzyme (or RNAP core enzyme) molecules and incubated for 5 min; wells were then washed with KG7 and movies were recorded. Next, observation wells containing RNAP-promoter complexes were supplemented with KG7 containing 250 µg/ml heparin (Sigma-Aldrich) solution, incubated for 1 min; wells were then washed with KG7 and movies were recorded.

For experiments monitoring RNAP-promoter complex formation reactions in real time (*Figures 1 and 2*, *Figure 1—figure supplement 3*, and *Figure 2—figure supplement 1*), observation wells containing immobilised RNAP were supplemented with KG7 imaging buffer, recordings were started, and Cy3-labelled promoter (or non-promoter) fragments were manually added (using a pipette) to the observation wells during the recording so as to yield a final concentration of 2 nM of Cy3-labelled

DNA in the wells. For experiments in *Figure 2—figure supplement 1A-C*, same procedures were followed, except that KG7 imaging buffer was supplemented with 20 µM Myx.

For experiments in *Figure 1—figure supplement 3B* top panel, observation wells containing immobilised RNAP-promoter complexes (formed with LC-[+ 2-Cy3]; sequence in *Figure 1—figure supplement 1*) were supplemented with KG7 buffer containing an NTP subset (500 µM ApU and 100 µM each of ATP, GTP, and UTP) directing synthesis of RNA up to position +11(RPitc ≤11), incubated for 5 min at 22 °C, solutions were removed, wells were washed with 2 × 30 µl KG7, and 30 µl KG7 imaging buffer at 22 °C was added. Movies of complexes corresponding to stalled RPitc ≤11 molecules were recorded. For experiments in *Figure 1—figure supplement 3B* bottom panel, observation wells containing immobilised stalled RPitc ≤11 molecules were supplemented with KG7 imaging buffer, the recording of a movie was started, and an NTP subset (100 µM each of GTP and CTP) directing extension of stalled complexes with 11-mer RNA to a 14-mer RNA was added in real time. Extension of RNA up to +11 results in stalled initial transcribing complexes (*Dulin et al., 2018*) and extension of RNA to +14 results in promoter escape and leads to formation of stalled elongation complexes for the lacCONS promoter used (Wang, Mazumder et al., in preparation).

## Single-molecule fluorescence microscopy: smFRET experiments

For experiments in *Figure 3* and *Figure 3—figure supplements 1–4*, observation wells of biotin-PEG-passivated glass surface were prepared and functionalised with Neutravidin (Sigma-Aldrich) to yield Neutravidin-biotin-PEG-functionalised glass floors. Biotin-tagged promoter DNA fragments were then immobilised in these observation wells as follows: aliquots (30 µl) of 0.05 nM biotin-tagged promoter fragments in KG7 were added to the observation chamber and incubated 1 min at 22 °C, solutions were removed, wells were washed with 2 × 30 µl KG7, and 30 µl KG7 imaging buffer at 22 °C was added.

For experiments in *Figure 3—figure supplement 1*, clamp-labelled RNAP molecules were manually added (final concentration of 2 nM) to the observation wells containing immobilised promoter DNA fragments, incubated for 5 min, wells were washed with KG7, and movies were recorded. Next, observation wells containing RNAP-promoter complexes were supplemented with KG7 containing 250 µg/ml heparin solution, incubated for 1 min, wells were washed with KG7, and movies were recorded.

For experiments monitoring RNAP-promoter complex formation reactions in real time (*Figure 3* and *Figure 3—figure supplements 2–4*), observation wells containing immobilised promoter (or non-promoter) fragments were supplemented with KG7 imaging buffer, recordings were started, and labelled-RNAP molecules (with Cy3B and Alexa647 at positions 284 on the β' subunit, and 106 on the β subunit of RNAP σ70 holoenzyme) were manually added (using a pipette) to the observation wells during the recording so as to yield a final concentration of 2 nM of labelled RNAP in the wells.

## Single-molecule fluorescence microscopy: data collection

Single-molecule fluorescence experiments were performed using a custom-built objective-type TIRF microscope (*Holden et al., 2010*). Light from a green laser (532 nm; Samba; Cobolt) and a red laser (635 nm; CUBE 635-30E, Coherent) was combined using a dichroic mirror coupled into a fiber-optic cable focused onto the rear focal plane of a 100 × oil-immersion objective (numerical aperture 1.4; Olympus) and was displaced off the optical axis, such that the incident angle at the oil-glass interface of a stage-mounted observation chamber exceeded the critical angle, thereby creating an exponentially decaying evanescent wave (*Axelrod et al., 1983*). Alternating-laser excitation (ALEX; *Kapanidis et al., 2006*; *Lee et al., 2005*) was implemented by directly modulating the green and red lasers using an acousto-optical modulator (1205 C, Isomet). Fluorescence emission was collected from the objective, was separated from excitation light using a dichroic mirror (545 nm/650 nm, Semrock) and emission filters (545 nm LP, Chroma; and 633/25 nm notch filter, Semrock), was focused on a slit to crop the image, and then was spectrally separated (using a dichroic mirror; 630 nm DLRP, Omega) into donor and emission channels focused side-by-side onto an electron-multiplying charge-coupled device camera (EMCCD; iXon 897; Andor Technology). A motorised x/y-scanning stage (MS-2000; ASI) was used to control the sample position relative to the objective.

For experiments in *Figures 1–2*, *Figure 1—figure supplements 2–4*, and *Figure 2—figure supplement 1*, frame rates were either 50 or 200 ms, and laser powers were either 0.60 or 0.15 mW

at 532 nm. For experiments in *Figure 3* and *Figure 3—figure supplements 2–3*, frames were either 100 or 200 ms long, and laser powers were either 200 µW in red, 500 µW in green, or 80 µW in red, 200 µW in green, respectively. For experiments in *Figure 3—figure supplement 4*, frame rates were 400 ms long, and laser powers were 50 µW in red, 150 µW in green. All data acquisition was carried out at 22 °C.

## Single-molecule fluorescence microscopy: data analysis for smUIFE experiments

For experiments in *Figures 1–2*, *Figure 1—figure supplements 2–4*, and *Figure 2—figure supplement 1*, localisation of single fluorescence emitters was detected and background-corrected fluorescence intensity vs. time trajectories from each localisation were extracted and curated to exclude trajectories exhibiting very high fluorescence intensities ($I_{Cy3}$) upon binding ($I_{Cy3}$ >750 photon counts), trajectories exhibiting multiple binding events during the observation window and trajectories exhibiting photoblinking. Initial inspection of time trajectories for experiments with lacCONS-[+ 2 Cy3] revealed three main classes of molecules. Class-I molecules started with the appearance of signal having intensity of ~200 photon counts, which remained stable until its disappearance (~46 %; *Figure 1—figure supplement 3*, *top*). Class-II molecules also started with appearance of a signal having intensity of ~200 photon counts, followed by intensity increase to ~450 photon counts after some time, followed by either signal disappearance or intensity decrease to ~200 photon counts (~45%: *Figure 1B*, *Figure 1—figure supplement 3*, *middle*). Class-III molecules started with appearance of a signal having intensity of ~450 photon counts, followed by intensity decrease to ~220 photon counts, and subsequent signal disappearance, or intensity increase to ~450 photon counts (~9 %; *Figure 1—figure supplement 3*, *bottom*). Similar observations were made for all other Cy3-labelled promoters studied.

Based on intensity levels, we assigned states with no signal to absence of a promoter DNA or to a bleached probe; states with ~200 photon counts to a promoter DNA that is dsDNA in the Cy3 vicinity; and states with ~450 counts to a promoter that is unwound in the Cy3 vicinity. We thus assigned events in Class-I molecules to binding of a promoter DNA molecule to RNAP, followed by dissociation of complexes, indicating formation of non-specific complexes; these events are also consistent with bleaching occurring prior to any intensity increase. We assigned events in Class-II molecules to binding of a promoter DNA molecule to RNAP, followed by promoter unwinding, followed by bleaching or promoter rewinding. Finally, we assigned events in Class-III molecules to binding of an unwound-promoter DNA molecule to RNAP, followed by promoter rewinding, followed by bleaching or promoter unwinding, indicating formation of complexes where the initial unwinding event is missed. We further curated intensity time trajectories up to the point of initial ~2 fold intensity enhancement for Class-II molecules only, since they showed an unambiguous bubble-opening event after initial binding.

Next, photon counts ($I_{Cy3}$) for the set of curated intensity vs. time trajectories were divided by 1000 to obtain $I_{Cy3}$*; the corresponding $I_{Cy3}$* vs. time trajectories were analysed to identify the different fluorescence intensity states using HMM as implemented in the MATLAB (MathWorks) software package ebFRET (*van de Meent et al., 2014*) using a three-state model (one of which corresponds to a negligible $I_{Cy3}$* state). The intensity $I_{Cy3}$*, for each of the other two HMM-derived states, was extracted from ebFRET and converted to $I_{Cy3}$ values, which in turn were binned and plotted as $I_{Cy3}$ count histograms (*Figure 1—figure supplement 4*, *Figure 2—figure supplement 1C*). These histograms were fitted using Gaussian distributions in Origin to define the mean fluorescence intensity that corresponds to each state (*Figure 1—figure supplement 4*, *Figure 2—figure supplement 1*).

Dwell times for each fluorescence intensity states were extracted from HMM fits to $I_{Cy3}$* vs. time trajectories, were binned, and were plotted as distribution histograms in Origin. For experiments in *Figures 1–2*, dwell-time distributions corresponding to the pre-unwinding state resembled peaked distributions indicating presence of at least two sub-steps and were fit to a two-exponential function of the form $y = A*(e^{-x/t1} - e^{-x/t2})$, where y represents counts of dwells in the closed-bubble state following the initial binding event and before the signal increase event, x represents time, and $t_1$ and $t_2$ represent the lifetime of the individual sub-steps. From the fit, we estimate the lifetimes corresponding to the two sub-steps, as well as the total time spent in the first closed-bubble state (as the sum of time spent in the two sub-steps; *Figure 2B*, right panels).

We note that it is possible that the transition involves more than two steps, but our current analysis and available data sets cannot specify the number and duration of each sub-step; consequently, we interpret qualitatively the shape of the dwell-time histogram as an indication of a multi-step process, and focus on the total time spent before the intensity enhancement event, since that can be interpreted with certainty as the time taken before unwinding in the vicinity of Cy3.

## Single-molecule fluorescence microscopy: data analysis for smFRET experiments

For experiments in *Figure 3* and *Figure 3—figure supplements 1–4*, localisations in donor-emission (green) and acceptor-emission (red) channels were detected using the peak-finding algorithm of the MATLAB (MathWorks) software package Twotone, as described (*Holden et al., 2010*). Peaks detected in both emission channels (i.e., peaks for molecules containing both donor and acceptor probes) were fitted with two-dimensional Gaussian functions to extract background-corrected intensity vs. time trajectories for donor-emission intensity upon donor excitation ($I_{DD}$), acceptor-emission intensity upon donor excitation ($I_{DA}$), and acceptor-emission intensity upon acceptor excitation ($I_{AA}$), as described (*Holden et al., 2010*). Intensity vs. time trajectories were curated to exclude trajectories exhibiting $I_{DD}$ <100 or > 1000 counts or $I_{AA}$ <200 or > 1000 counts, trajectories exhibiting multiple-step donor or acceptor photobleaching, trajectories exhibiting donor or acceptor photo-blinking and portions of trajectories following donor or acceptor photobleaching. Intensity vs. time trajectories were used to calculate trajectories of apparent donor-acceptor smFRET efficiency (E*) as described (*Holden et al., 2010*; *Kapanidis et al., 2004*; *Lee et al., 2005*):

E* = $I_{DA}$/ ($I_{DD}$ + $I_{DA}$)

E* time trajectories were analysed globally to identify E* states by use of HMM as implemented in MATLAB (MathWorks) software package ebFRET (*van de Meent et al., 2014*).

HMM analysis of E* time trajectories revealed four types of molecules. The most abundant were Class-I molecules (~45 % of all events) that started with E*~0.40, remained at E*~0.40, and then bleached (*Figure 3—figure supplement 3A*), and Class-II molecules (~44%) that started with E*~0.40, transitioned to E*~0.48, and then bleached or returned to E*~0.40 (*Figure 3—figure supplement 3B*, *top*). More rarely, we observed Class-III molecules (~9 % of all events) that started with E*~0.48, transitioned to E*~0.40, followed by bleaching or transition to E*~0.48; *Figure 3—figure supplement 3C*, and, very rarely, we observed Class-IV molecules (~2%) that started with E*~0.40, switched between E*~0.40–0.20 and bleached (*Figure 3—figure supplement 3D*).

We refer to the new E*~0.48 state observed here as the 'locked-clamp' conformation, with Class-II molecules providing information on the time taken to form the 'locked-clamp' conformation; in contrast, Class-I molecules represent RNAP molecules that either bleach before the transition to the locked-clamp state or bind non-specifically to DNA; and Class-III molecules represent molecules where the initial clamp locking is missed.

E* time trajectories for Class-II molecules were fitted to a two-state HMM model, E*-values from the fitted model were extracted, binned, and plotted using Origin (Origin Lab), and were fitted to Gaussian distributions using Origin (*Figure 3B*, right panels and *Figure 3—figure supplement 3B*, left panel; coloured curves). The resulting histograms provide population distributions of E* states and, for each E* state, define mean E* (*Figure 3B*, right panels and *Figure 3—figure supplement 4B*, right panel; coloured bars and inset). Dwell-time distributions corresponding to time spent before the first transition from a closed-clamp state to a locked-clamp state were extracted from HMM fits to E* vs. time trajectories, and were binned and plotted as distribution histograms in Origin (*Figure 3* and *Figure 3—figure supplement 4C*). Dwell-time histograms obtained in this manner exhibited the shape of a peaked distribution, indicating presence of at least two sub-steps, were fitted to a two-exponential function of the form y = A*($e^{-x/t1}$ – $e^{-x/t2}$), where y represents counts of dwells in the closed-clamp state following the initial binding event and before the clamp-locking event, x represents time, and $t_1$ and $t_2$ represent the lifetimes of the individual sub-steps. From the fit, we estimate the time spent in the two sub-steps, as well as the total time before transition to 'locked-clamp' conformation ($t_{LOCK}$) from the sum of times corresponding to the two sub-steps. Similar to the case for fluorescence enhancement experiments described previously, we note that there may be more than two steps which contribute significantly to these dwell times, and we could not infer an accurate model in terms of the number and duration of each sub-step involved from this dataset. Therefore, we avoid

assigning the two sub-steps to specific conformations or events and focus on the total time spent before the transition to 'locked-clamp' conformation.

## Data and software availability

All information for replication is included in the submission and data corresponding to each figure are provided as source data files. MATLAB software packages TwoTone and ebFRET are available on GitHub (https://github.com/annawang692/TwoTone2018, *Wang, 2020* and http://ebfret.github.io/).

## Acknowledgements

We thank Dr Anssi Malinen for discussions and early work on the development of real-time smFRET assays, and Dr Horst Steuer for the development of custom software. This work was supported by the Wellcome Trust [110164/Z/15/Z to ANK] and NIH [GM041376 to RHE].

The illustrations in Figures 1A, 2A, 3A, and 4 were created using Biorender (Biorender.com).

# Additional information

### Funding

| Funder | Grant reference number | Author |
| --- | --- | --- |
| Wellcome Trust | 110164/Z/15/Z | Achillefs Kapanidis |
| NIH Office of the Director | GM041376 | Richard H Ebright |

The funders had no role in study design, data collection and interpretation, or the decision to submit the work for publication.

### Author contributions

Abhishek Mazumder, Conceptualization, Formal analysis, Funding acquisition, Supervision, Writing – original draft, Investigation, Methodology; Richard H Ebright, Conceptualization, Resources, Writing – review and editing, Methodology; Achillefs N Kapanidis, Conceptualization, Funding acquisition, Resources, Writing – review and editing, Data curation, Investigation, Methodology

### Author ORCIDs

Abhishek Mazumder http://orcid.org/0000-0002-9339-6256
Richard H Ebright http://orcid.org/0000-0001-8915-7140
Achillefs N Kapanidis http://orcid.org/0000-0001-6699-136X

### Decision letter and Author response

Decision letter https://doi.org/10.7554/eLife.70090.sa1
Author response https://doi.org/10.7554/eLife.70090.sa2

# Additional files

### Supplementary files

• Transparent reporting form

### Data availability

All information for replication is included in the submission and data corresponding to each figure are provided as source data files. MATLAB software packages TwoTone and ebFRET are available on Github (https://github.com/annawang692/TwoTone2018 and http://ebfret.github.io/).

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
