## [Decision Letter]

**Acceptance summary:**

This work provides new insights into the molecular mechanism by which RNA polymerase separates the two strands of DNA, generating a single-stranded template for RNA synthesis. Using single-molecule analysis the authors examined two conformational transitions taking place during RNA transcription initiation: DNA unwinding and RNAP clamp movements. The paper will help to distinguish between two competing hypotheses within the literature. Namely, using the bacterial consensus promoter the authors showed that a clamp-opening event is not obligatory for promoter unwinding and that promoter unwinding proceeds via a "bind-unwind-load-and-lock" model. The work will be of relevance to a wide range of researchers interested in the molecular basis of gene expression and gene regulation.

**Decision letter after peer review:**

Thank you for submitting your article "Transcription initiation at a consensus bacterial promoter proceeds via a "bind-unwind-load- and-lock" mechanism" for consideration by *eLife*. Your article has been reviewed by 2 peer reviewers, and the evaluation has been overseen by Maria Spies as a Reviewing Editor and Cynthia Wolberger as the Senior Editor. The following individual involved in review of your submission has agreed to reveal their identity: Nigel J Savery (Reviewer #2).

Essential revisions:

The reviewers and the reviewing editor found the work well executed and technically sounds, but concluded that additional experiments are needed to exclude the model where the RNAP clamp needs to acquire more open conformations in order for DNA unwinding to take place.

1. Controls suggested by reviewer 1, points (1) and (2) are essential.

2. The results currently presented in the paper are for a consensus promoter. Including another promoter or two would constitute reasonable amount of work and would greatly increase the generality and impact of the work. Studying multiple promoters within the context of one system and one group seems the best way for deciding if the same mechanism is always observed or if a variety of different mechanisms is possible (which may explain the existence of support for different mechanisms in the field).

3. The Discussion is quite brief. The authors could discuss their results in the context of the evidences for other models. In particular, the issue of alternative means through which the DNA may be loaded in the cleft despite a lack of conformational change in the clamp should be discussed.

4. The proposition that no "open" clamp conformations were observed in the E* traces needs further discussion. Considering the time resolution of the experiments, very rapid opening would not be detected. Additionally, DNA opening and RNA polymerase opening were measured in separate experiments and therefore, the conclusions rely in part on correlation between results gained in slightly different conditions. This needs to be discussed.

*Reviewer #1 (Recommendations for the authors):*

(1) The claim by the authors that heparin-resistance shows specificity is not convincing as heparin as a variety of effects depending on the system. The authors should do not provide traces from Cy3-labeled promoter-less DNA templates as negative controls. Nor do they provide traces using promoter-less templates to recruit labeled RNAP to the surface. These experiments would allow for the facile identification of non-specific signals. In addition, on promoter templates, core RNAP in the absence of σ factor could be used as a negative control for promoter-specific signals.

(2) No NTP chase controls are shown to suggest that any of these complexes are on-pathway and capable of escaping the promoter.

(3) These experiments are done with the lacCONS promoter which does not faithfully represent real *E. coli* promoters. In fact, it is rare for real promoters to have fully consensus sequences. In addition, there may be multiple conformational pathways for DNA unwinding during initiation. Without measuring the process using a variety of real promoters taken from *E. coli*, no general conclusions regarding "the" pathway of initiation can be made.

(4) The premise is that no "open" clamp conformations are observed in the E* traces. However, I suppose that only a lower limit on the lifetime of such a state can be given based on the time resolution of the measurement.

(5) It is not clear how to map "openness" from FRET measurements presented here to those made in structural work unless there is some calibrated way to convert E* to a distance.

The combination of these points really limits the impact that this manuscript would have on the field.

*Reviewer #2 (Recommendations for the authors):*

The manuscript was presented logically and clearly and I have only a few suggestions for the authors to consider.

1. The smFRET experiments directly examine the state of the clamp, and the experiments support the conclusion that the labelled residues do not adopt the "open clamp" conformation during promoter melting (within the time resolution of the experiments). A recent paper has suggested that movement of the Β lobe might allow loading of double-stranded DNA even in the presence of what they called a "partially closed" clamp (Unarta et al., PNAS 2021 PNAS April 27, 2021 118 (17) e2024324118). Ref 17 in the current ms, which examined open complex formation in a different system using cryo-em, also raised the possible importance of movements of the Β lobe for loading of dsDNA. These findings do not affect the validity of the author's conclusions about the state of the clamp in their system, but do suggest that some caution might be needed when deciding how strongly these observations about the clamp suggest that unwinding must be taking place prior to loading. It would be useful to discuss the potential role of RNA polymerase conformational changes that would not be reflected in the smFRET analysis when interpreting the data.

2. P8 Para 2 line 3. The term E* is introduced for the first time in the manuscript here and it would be helpful to provide more explanation for readers not familiar with the technique.

3. P8 It is shown that Myx prevents the formation of heparin-resistant complexes, but the reason why the open complexes are not heparin-resistant is not clear, given that promoter unwinding occurs. It would be helpful to add more explanation here: do the authors think that Myx prevents the "lock" stage?

4. Figure 2B: is the very short lifetime of the unwound state in the example shown for the +2 construct typical of the data collected and was the lifetime of the -9 constructs typically longer? If so, does this indicate that the presence of the probe at that position is affecting the stability of the open complex (and hence potentially the kinetics of unwinding?) I note that this is related in part to point 3 as these experiments are done with Myx, and so the complexes are heparin-sensitive.

5. Figure S2B, Figure S5B, Figure S6B: number of repeats is not stated.

---

## [Author Response]

Essential revisions:The reviewers and the reviewing editor found the work well executed and technically sounds, but concluded that additional experiments are needed to exclude the model where the RNAP clamp needs to acquire more open conformations in order for DNA unwinding to take place.1. Controls suggested by reviewer 1, points (1) and (2) are essential.

We have now added results from these experiments; please see answers to Reviewer 1.

2. The results currently presented in the paper are for a consensus promoter. Including another promoter or two would constitute reasonable amount of work and would greatly increase the generality and impact of the work. Studying multiple promoters within the context of one system and one group seems the best way for deciding if the same mechanism is always observed or if a variety of different mechanisms is possible (which may explain the existence of support for different mechanisms in the field).

We agree with the editor and reviewer on this general point and have clarified this point in the answer to Reviewer 1.

3. The Discussion is quite brief. The authors could discuss their results in the context of the evidences for other models. In particular, the issue of alternative means through which the DNA may be loaded in the cleft despite a lack of conformational change in the clamp should be discussed.

We have modified the discussion to include a discussion of the potential importance of β-lobe conformational changes, and of the importance of studying different promoters to infer the general and specific features of promoter unwinding in transcription initiation. See responses to reviewer 1 (query no. 3) and reviewer 2 (query no. 1).

4. The proposition that no "open" clamp conformations were observed in the E* traces needs further discussion. Considering the time resolution of the experiments, very rapid opening would not be detected.

The point about limits set by our time resolution was raised by Reviewer 1; please see our reply to query no. 4.

Additionally, DNA opening and RNA polymerase opening were measured in separate experiments and therefore, the conclusions rely in part on correlation between results gained in slightly different conditions. This needs to be discussed.

We have added the following in the discussion:

“Extension of these assays using more complex labelling and imaging schemes should allow simultaneous observation of promoter unwinding and RNAP clamp conformational changes in the same molecule, enabling a more detailed view of the coupling of the molecular motions involved in promoter unwinding.”

With the new control experiments, changes in the text, and responses to both reviewers, we believe that we have addressed the main concerns of the reviewers, and we believe that our work will have significant impact in the field.

Reviewer #1 (Recommendations for the authors):(1) The claim by the authors that heparin-resistance shows specificity is not convincing as heparin as a variety of effects depending on the system. The authors should do not provide traces from Cy3-labeled promoter-less DNA templates as negative controls. Nor do they provide traces using promoter-less templates to recruit labeled RNAP to the surface. These experiments would allow for the facile identification of non-specific signals. In addition, on promoter templates, core RNAP in the absence of σ factor could be used as a negative control for promoter-specific signals.

We have now included results of the control experiments suggested by the reviewer: (i) experiments using a Cy3-labelled non-promoter DNA fragment and immobilised RNAP holoenzyme, (ii) experiments using a Cy3-labelled promoter DNA fragment and immobilised RNAP core enzyme, and (iii) experiments using a doubly labelled RNAP holoenzyme and an immobilised non-promoter DNA fragment. The results show clearly that the fluorescence enhancement signals observed using RNAP holoenzyme and promoter DNA fragments are specific to promoter-unwinding events.

We have added the following text describing these control experiments:

“Control experiments show that: (i) Cy3-containing promoter DNA fragments bound exclusively to immobilised RNAP molecules (Figure S2A), and the majority (~60%) of the resulting complexes were resistant to challenge with heparin (Figure S2B); (ii) binding of Cy3-containing non-promoter DNA fragments to immobilised RNAP holoenzyme molecules did not result in heparin-resistant complexes (Figure S2C); and (ii) binding of Cy3-containing promoter DNA fragments to immobilised RNAP core enzyme molecules did not result in heparin-resistant complexes (Figure S2D). These results confirm that sequence-specific, heparin-resistant, complexes were formed only between RNAP holoenzyme and Cy3-containing promoter DNA fragments.”

and,

“Control experiments show that (i) binding of Cy3-containing non-promoter DNA fragments to immobilised RNAP holoenzyme did not result in ≥2-fold fluorescence enhancement (Figure S3C); and (ii) binding of Cy3-containing promoter DNA fragments to immobilised RNAP core molecules did not result in ≥2-fold fluorescence enhancement events for the overwhelming majority of the time-trajectories (~96%; Figure S3D). These results indicate that the detected ~2-fold fluorescence enhancement events represent sequence-specific complexes between RNAP holoenzyme and promoter DNA fragments.”

and,

“Control experiments show that: (i) doubly-labelled RNAP molecules bound exclusively to immobilised promoter DNA molecules (Figure S6A); (ii) the majority (~87%) of the resulting complexes were resistant to challenge with heparin (Figure S6B); and (iii) doubly-labelled RNAP molecules did not bind stably to immobilised non-promoter DNA molecules (Figure S7C). These results indicate sequence-specific, heparin-resistant RNAP-promoter complexes were only formed between the doubly labelled RNAP holoenzyme molecules and promoter DNA fragments.”

(2) No NTP chase controls are shown to suggest that any of these complexes are on-pathway and capable of escaping the promoter.

We have now included NTP chase control experiments showing that complexes formed in these experiments are transcription competent and capable of escaping the promoter. We have added the following text:

“Upon addition of an NTP subset enabling synthesis of transcripts up to 11-nt in length (ATP, UTP and GTP), ~49% of trajectories showed formation of a stable unwound-promoter state (with fluorescence intensity similar to that of the sub-population exhibiting promoter unwinding), as expected for the formation of initial transcribing complexes (RPitc <11; Figure S3B, top; see also Methods). Upon subsequent addition of a separate NTP subset (GTP and CTP), enabling synthesis of transcripts up to 14-nt in length, ~27% of the resulting complexes showed formation of a stable rewound-promoter state (Figure S3B, bottom), as expected for promoter escape and formation of a transcription elongation complex. The results show that RNAP-promoter complexes formed in these experiments are transcriptionally active and are competent for promoter escape (see also Methods).”

Further, we have added the following text to Methods:

“For experiments in Figure S3B top panel, observation wells containing immobilised RNAP-promoter complexes (formed with LC-[+2-Cy3]; sequence in Figure S1) were supplemented with KG7 buffer containing a NTP subset (500 µM ApA and 100 µM each of ATP, GTP and UTP) directing synthesis of RNA up to position +11(RPitc <11), incubated for 5 min at 22ºC, solutions were removed, wells were washed with 2 x 30 µl KG7, and 30 µl KG7 imaging buffer at 22ºC was added. Videos of complexes corresponding to stalled RPitc <11 molecules were recorded. For experiments in Figure S3B bottom panel, observation wells containing immobilised stalled RPitc <11 molecules were supplemented with KG7 imaging buffer, the recording of a video was started, and an NTP subset (100 µM each of GTP and CTP) directing extension of stalled complexes with 11-mer RNA to a 14-mer RNA, was added in real time. Extension of RNA up to +11 results in stalled initial transcribing complexes (45) and extension of RNA to +14 results in promoter escape and leads to formation of stalled elongation complexes for the lacCONS promoter used (Wang, Mazumder et al., in preparation).”

(3) These experiments are done with the lacCONS promoter which does not faithfully represent real E. coli promoters. In fact, it is rare for real promoters to have fully consensus sequences. In addition, there may be multiple conformational pathways for DNA unwinding during initiation. Without measuring the process using a variety of real promoters taken from *E. coli*, no general conclusions regarding "the" pathway of initiation can be made.

We thank the reviewer for raising this important point. As the reviewer mentions, and as we state clearly in our manuscript, we have performed our experiments using *lacCONS*, a derivative of the *lac* promoter derivative that includes a consensus σ70 bacterial promoter. For that promoter, as well as for several derivatives thereof, we show that a clamp-opening event is not obligatory for promoter unwinding and that promoter unwinding proceeds via a “bind-unwind-load-and-lock” model. We point out that the *lacCONS* promoter has been used extensively in previous work to establish transcription mechanisms: e.g., clamp closure upon RNAP-promoter open-complex formation (Chakraborty et al., *Science* 2012), DNA scrunching in transcription-start-site selection (Winkelman et al., *Science* 2016), DNA scrunching in initial transcription (Kapanidis et al., *Science* 2006), and RNAP pausing in initial transcription (Duchi et al., *Mol Cell* 2016: Winkelman et al., *Mol Cell* 2020), all of which subsequently were validated for other promoters.

Nevertheless, we agree with the reviewer that, within the general framework of a “bind-unwind-load-and-lock” model, different promoter unwinding kinetics and pathways are possible for promoters with different sequence elements. Being aware of this issue, we have clearly stated (both in the title and throughout the manuscript) that our study specifically concerns a consensus bacterial promoter.

In the discussion we now write:

“The consensus bacterial promoter employed in this study has been used in previous work to define transcription mechanisms, including RNAP clamp closure upon RPo formation (14), DNA scrunching in transcription-start-site selection (40), DNA scrunching in initial transcription (41), and RNAP pausing in initial transcription (42,43), all of which subsequently were validated for other promoters. Nevertheless, we emphasize that it remains to be determined whether the bind-unwind-load-and-lock pathway defined here for the consensus bacterial promoter also is used by E. coli RNAP σ^70^ holoenzyme at other promoters and by other *E. coli* RNAP holoenzymes.”

We have begun extending our work--focusing mainly on the kinetics of promoter unwinding, rather than on RNAP clamp conformation--to other promoters. However, analysis of individual other promoters will not enable the question to be settled in a comprehensive and conclusive manner. Therefore, in further separate work, we plan to develop and apply massively parallel single-molecule assays that interrogate large libraries of promoters. This body of work will require time and clearly is separate from the present body of work.

We have also added the following text to the Discussion:

“Finally, development of massively parallel single-molecule assays interrogating large libraries of promoter sequences, would enable a comprehensive analysis of general and specific features of this first step in transcription initiation for different promoter sequences.”

(4) The premise is that no "open" clamp conformations are observed in the E* traces. However, I suppose that only a lower limit on the lifetime of such a state can be given based on the time resolution of the measurement.

The reviewer is correct that, in principle, an obligatory clamp-opening/clamp-reclosing event during RPo formation possibly could be missed by our smFRET experiments, if it occurred on a time scale faster than the ~100 ms temporal resolution of our smFRET experiments. However, we point out that the possibility of an obligatory clamp-opening/clamp-reclosing event occurring on *any* time scale during RPo formation is excluded by our smUIFE experiments showing that the RNAP inhibitor myxopyronin, which locks the RNAP clamp in the closed state, does not block, or even delay, promoter unwinding (Figure 2).

This following text makes this point:

"To determine whether promoter unwinding is affected by preventing opening of the RNAP clamp, we repeated our experiments in presence of myxopyronin (Myx; 14, 15, 24, 37-38), an RNAP inhibitor that prevents RNAP clamp opening (14, 24, 37; state with an smFRET efficiency, E*~0.36 in Figure S5D) and that allows formation of heparin-sensitive RNAP-promoter complexes but prevents formation of heparin-resistant RNAP-promoter complexes (37; Figures S5A-B). […] We infer--contrary to the models in which clamp opening is obligatory for promoter unwinding--that preventing RNAP clamp opening does not prevent promoter unwinding, and increases, not decreases, the kinetics of promoter unwinding."

(5) It is not clear how to map "openness" from FRET measurements presented here to those made in structural work unless there is some calibrated way to convert E* to a distance.

The FRET construct used here for monitoring clamp conformation has been characterised and compared to structural work in previous studies (Chakraborty et al., Science, 2012; Duchi et al., NAR, 2018; Mazumder et al., NAR, 2021). We precisely mapped the three FRET states centred on ~0.2, ~0.3 and ~0.4 to open, partly closed and closed structural states respectively, from probe accessible volume calculations on available structures and distance estimations from accurate FRET values in Duchi et al., NAR, 2018.

The combination of these points really limits the impact that this manuscript would have on the field.Reviewer #2 (Recommendations for the authors):The manuscript was presented logically and clearly and I have only a few suggestions for the authors to consider.1. The smFRET experiments directly examine the state of the clamp, and the experiments support the conclusion that the labelled residues do not adopt the "open clamp" conformation during promoter melting (within the time resolution of the experiments). A recent paper has suggested that movement of the Β lobe might allow loading of double-stranded DNA even in the presence of what they called a "partially closed" clamp (Unarta et al., PNAS 2021 PNAS April 27, 2021 118 (17) e2024324118). Ref 17 in the current ms, which examined open complex formation in a different system using cryo-em, also raised the possible importance of movements of the Β lobe for loading of dsDNA. These findings do not affect the validity of the author's conclusions about the state of the clamp in their system, but do suggest that some caution might be needed when deciding how strongly these observations about the clamp suggest that unwinding must be taking place prior to loading. It would be useful to discuss the potential role of RNA polymerase conformational changes that would not be reflected in the smFRET analysis when interpreting the data.

The reviewer is correct that a molecular-dynamics simulation has raised the possibility that motions of the RNAP β lobe, termed "gate opening," could make part of the RNAP active-centre cleft accessible to dsDNA without RNAP clamp opening. We have added the following text in the Discussion:

“As described in the introduction, it has been the predominant view that structural considerations necessitate RNAP clamp opening for loading of dsDNA into the RNAP active-centre cleft but do not necessitate RNAP clamp opening for loading of ssDNA into the RNAP active-centre cleft (1-2,7-9,11,13-14). Under that predominant view, our finding that the RNAP clamp remains closed throughout promoter unwinding at a consensus bacterial promoter implies that it is ssDNA, not dsDNA, that enters the RNAP active-centre-cleft, and thus that promoter unwinding occurs at least in part outside, not inside, the RNAP active-centre cleft (Figure 4).

In contrast to the predominant view, it recently has been speculated, based on molecular-dynamics simulations, that part of the RNAP cleft potentially could become accessible to both dsDNA and ssDNA without clamp opening, due to motions, putatively occurring on the nanosecond time-scale, of the RNAP β lobe ("gate opening"; 39). Under this alternative view, our finding that the RNAP clamp remains closed throughout promoter unwinding at a consensus bacterial promoter potentially could be consistent with either ssDNA or dsDNA entering the RNAP active-centre-cleft, and thus potentially could be consistent with promoter unwinding that occurs either outside or inside, the RNAP active-centre cleft. Nevertheless, in view of the absence of experimental support for the alternative view, and in view of the mismatch between the nanosecond time scale of the limiting reaction in the alternative view and the millisecond-to-second time scales of promoter unwinding, we tend to disfavour this alternative view.”

2. P8 Para 2 line 3. The term E* is introduced for the first time in the manuscript here and it would be helpful to provide more explanation for readers not familiar with the technique.

We have replaced "E*" by “an apparent smFRET efficiency, E*”.

3. P8 It is shown that Myx prevents the formation of heparin-resistant complexes, but the reason why the open complexes are not heparin-resistant is not clear, given that promoter unwinding occurs. It would be helpful to add more explanation here: do the authors think that Myx prevents the "lock" stage?

We suggest that Myx prevents formation of heparin-resistant open complexes either by preventing loading of unwound DNA into active centre cleft or by preventing subsequent locking of the RNAP clamp. We have added the following clarification in the main text:

“We suggest that Myx prevents formation of heparin-resistant open complexes either by preventing loading of unwound DNA into active centre cleft or by preventing subsequent locking of the RNAP clamp.”

4. Figure 2B: is the very short lifetime of the unwound state in the example shown for the +2 construct typical of the data collected and was the lifetime of the -9 constructs typically longer? If so, does this indicate that the presence of the probe at that position is affecting the stability of the open complex (and hence potentially the kinetics of unwinding?) I note that this is related in part to point 3 as these experiments are done with Myx, and so the complexes are heparin-sensitive.

The lifetime of the unwound state in Figure 2B is not typical of the dataset. Further, we note that end of a time-trajectory signifies either an isomerisation to a rewound state, or bleaching and is not generally indicative of the lifetime in the state. In a separate study, we have been examining lifetimes of unwound states. In that study, the time-traces used are long enough to observe entire dwells in the unwound state and to observe that dwells in the unwound state end by transitions to other intensity states.

5. Figure S2B, Figure S5B, Figure S6B: number of repeats is not stated.

We have added this information in the respective figure legends.

References

Chakraborty, A., Wang, D., Ebright, Y.W., Korlann, Y., Kortkhonjia, E., Kim, T., Chowdhury, S., Wigneshweraraj, S., Irschik, H., Jansen, R., Nixon, B.T., Knight, J., Weiss, S., and Ebright, R.H. (2012) Opening and closing of the bacterial RNA polymerase clamp. *Science,* 337, 591-595.

Winkelman, J. T., Vvedenskaya, I. O., Zhang, Y., Zhang, Y., Bird, J. G., Taylor, D. M., Gourse, R. L., Ebright, R. H., and Nickels, B. E. (2016) Multiplexed protein-DNA cross-linking: Scrunching in transcription start site selection. *Science*, 351, 1090–1093.

Kapanidis, A. N., Margeat, E., Ho, S. O., Kortkhonjia, E., Weiss, S., and Ebright, R. H. (2006). Initial transcription by RNA polymerase proceeds through a DNA-scrunching mechanism. *Science*, 314, 1144–1147.

Duchi D, Bauer D.L., Fernandez L., Evans G., Robb N., Hwang L.C., Gryte K., Tomescu A., Zawadzki P., Morichaud Z., Brodolin K., Kapanidis A.N. (2016) RNA Polymerase Pausing during Initial Transcription. *Molecular cell*, 63, 939-50.

Winkelman, J. T., Pukhrambam, C., Vvedenskaya, I. O., Zhang, Y., Taylor, D. M., Shah, P., Ebright, R. H., and Nickels, B. E. (2020) XACT-Seq Comprehensively Defines the Promoter-Position and Promoter-Sequence Determinants for Initial-Transcription Pausing. *Molecular cell*, 79, 797–811.

Duchi, D., Mazumder, A., Malinen, A. M., Ebright, R. H., Kapanidis, A. N. (2018) The RNA polymerase clamp interconverts dynamically among three states and is stabilized in a partly closed state by ppGpp. *Nucleic Acids Res,* 46, 7284-7295.

Mazumder, A., Wang, A., Uhm, H., Ebright, R.H., Kapanidis, A.N. (2021) RNA polymerase clamp conformational dynamics: long-lived states and modulation by crowding, cations, and nonspecific DNA binding. *Nucleic Acids Res,* Advance online article.